# A simple neural network module
# for relational reasoning

**Adam Santoro**[*]
adamsantoro@google.com

**David Raposo**[*]
draposo@google.com

**David G.T. Barrett**
barrettdavid@google.com

**Mateusz Malinowski**
mateuszm@google.com

**Razvan Pascanu**
razp@google.com

**Peter Battaglia**
peterbattaglia@google.com

**Timothy Lillicrap**
DeepMind
London, United Kingdom
countzero@google.com

## Abstract

Relational reasoning is a central component of generally intelligent behavior, but has proven difficult for neural networks to learn. In this paper we describe how to use Relation Networks (RNs) as a simple plug-and-play module to solve problems that fundamentally hinge on relational reasoning. We tested RN-augmented networks on three tasks: visual question answering using a challenging dataset called CLEVR, on which we achieve state-of-the-art, super-human performance; text-based question answering using the bAbI suite of tasks; and complex reasoning about dynamic physical systems. Then, using a curated dataset called Sort-of-CLEVR we show that powerful convolutional networks do not have a general capacity to solve relational questions, but can gain this capacity when augmented with RNs. Thus, by simply augmenting convolutions, LSTMs, and MLPs with RNs, we can remove computational burden from network components that are not well-suited to handle relational reasoning, reduce overall network complexity, and gain a general ability to reason about the relations between entities and their properties.

## 1   Introduction

The ability to reason about the relations between entities and their properties is central to generally intelligent behavior (Figure 1) [10, 7]. Consider a child proposing a race between the two trees in the park that are furthest apart: the pairwise distances between every tree in the park must be inferred and compared to know where to run. Or, consider a reader piecing together evidence to predict the culprit in a murder-mystery novel: each clue must be considered in its broader context to build a plausible narrative and solve the mystery.

Symbolic approaches to artificial intelligence are inherently relational [16, 5]. Practitioners define the relations between symbols using the language of logic and mathematics, and then reason about these relations using a multitude of powerful methods, including deduction, arithmetic, and algebra. But symbolic approaches suffer from the symbol grounding problem and are not robust to small

---

[*]Equal contribution.

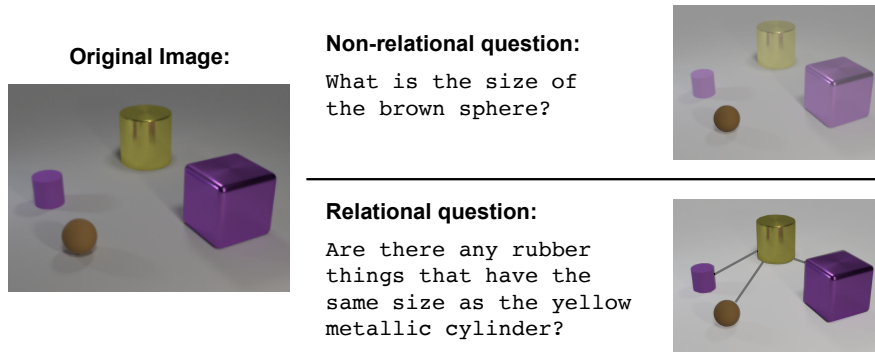

**Original Image:**

**Non-relational question:**

What is the size of
the brown sphere?

**Relational question:**

Are there any rubber
things that have the
same size as the yellow
metallic cylinder?

Figure 1: **An illustrative example from the CLEVR dataset of relational reasoning**. An image containing four objects is shown alongside non-relational and relational questions. The relational question requires explicit reasoning about the relations between the four objects in the image, whereas the non-relational question requires reasoning about the attributes of a particular object.

task and input variations [5]. Other approaches, such as those based on statistical learning, build representations from raw data and often generalize across diverse and noisy conditions [12]. However, a number of these approaches, such as deep learning, often struggle in data-poor problems where the underlying structure is characterized by sparse but complex relations [3, 11]. Our results corroborate these claims, and further demonstrate that seemingly simple relational inferences are remarkably difficult for powerful neural network architectures such as convolutional neural networks (CNNs) and multi-layer perceptrons (MLPs).

Here, we explore "Relation Networks" (RN) as a general solution to relational reasoning in neural networks. RNs are architectures whose computations focus explicitly on relational reasoning [18]. Although several other models supporting relation-centric computation have been proposed, such as Graph Neural Neworks, Gated Graph Sequence Neural Netoworks, and Interaction Networks, [20, 13, 2], RNs are simpler, more exclusively focused on general relation reasoning, and easier to integrate within broader architectures. Moreover, RNs require minimal oversight to construct their input, and can be applied successfully to tasks even when provided with relatively unstructured inputs coming from CNNs and LSTMs. We applied an RN-augmented architecture to CLEVR [7], a recent visual question answering (QA) dataset on which state-of-the-art approaches have struggled due to the demand for rich relational reasoning. Our networks vastly outperformed the best generally-applicable visual QA architectures, and achieve state-of-the-art, super-human performance. RNs also solve CLEVR from state descriptions, highlighting their versatility in regards to the form of their input. We also applied an RN-based architecture to the bAbI text-based QA suite [22] and solved 18/20 of the subtasks. Finally, we trained an RN to make challenging relational inferences about complex physical systems and motion capture data. The success of RNs across this set of substantially dissimilar task domains is testament to the general utility of RNs for solving problems that require relation reasoning.

## 2 Relation Networks

An RN is a neural network module with a structure primed for relational reasoning. The design philosophy behind RNs is to constrain the functional form of a neural network so that it captures the core common properties of relational reasoning. In other words, the capacity to compute relations is baked into the RN architecture without needing to be learned, just as the capacity to reason about spatial, translation invariant properties is built-in to CNNs, and the capacity to reason about sequential dependencies is built into recurrent neural networks.

In its simplest form the RN is a composite function:

$$\mathrm{RN}(O) = f_\phi \left( \sum_{i,j} g_\theta(o_i, o_j) \right),$$  (1)

where the input is a set of "objects" $O = \{o_1, o_2, ..., o_n\}$, $o_i \in \mathbb{R}^m$ is the $i^{th}$ object, and $f_\phi$ and $g_\theta$ are functions with parameters $\phi$ and $\theta$, respectively. For our purposes, $f_\phi$ and $g_\theta$ are MLPs, and the parameters are learnable synaptic weights, making RNs end-to-end differentiable. We call the output of $g_\theta$ a "relation"; therefore, the role of $g_\theta$ is to infer the ways in which two objects are related, or if they are even related at all.

RNs have three notable strengths: they learn to infer relations, they are data efficient, and they operate on a set of objects – a particularly general and versatile input format – in a manner that is order invariant.

**RNs learn to infer relations**  The functional form in Equation 1 dictates that an RN should consider the potential relations between *all* object pairs. This implies that an RN is not necessarily privy to which object relations actually exist, nor to the actual meaning of any particular relation. Thus, RNs must *learn to infer* the existence and implications of object relations.

In graph theory parlance, the input can be thought of as a complete and directed graph whose nodes are objects and whose edges denote the object pairs whose relations should be considered. Although we focus on this "all-to-all" version of the RN throughout this paper, this RN definition can be adjusted to consider only some object pairs. Similar to Interaction Networks [2], to which RNs are related, RNs can take as input a list of only those pairs that should be considered, if this information is available. This information could be explicit in the input data, or could perhaps be extracted by some upstream mechanism.

**RNs are data efficient**  RNs use a single function $g_\theta$ to compute each relation. This can be thought of as a single function operating on a batch of object pairs, where each member of the batch is a particular object-object pair from the same object set. This mode of operation encourages greater generalization for computing relations, since $g_\theta$ is encouraged not to over-fit to the features of any particular object pair. Consider how an MLP would learn the same function. An MLP would receive *all* objects from the object set simultaneously as its input. It must then learn and embed $n^2$ (where $n$ is the number of objects) identical functions within its weight parameters to account for all possible object pairings. This quickly becomes intractable as the number of objects grows. Therefore, the cost of learning a relation function $n^2$ times using a single feedforward pass per sample, as in an MLP, is replaced by the cost of $n^2$ feedforward passes per object set (i.e., for each possible object pair in the set) and learning a relation function just once, as in an RN.

**RNs operate on a set of objects**  The summation in Equation 1 ensures that the RN is invariant to the order of objects in its input, respecting the property that sets are order invariant. Although we used summation, other commutative operators – such as max, and average pooling – can be used instead.

## 3  Tasks

We applied RN-augmented networks to a variety of tasks that hinge on relational reasoning. To demonstrate the versatility of these networks we chose tasks from a number of different domains, including visual QA, text-based QA, and dynamic physical systems.

### 3.1  CLEVR

In visual QA a model must learn to answer questions about an image (Figure 1). This is a challenging problem domain because it requires high-level scene understanding [1, 14]. Architectures must perform complex relational reasoning – spatial and otherwise – over the features in the visual inputs, language inputs, and their conjunction. However, the majority of visual QA datasets require reasoning in the absence of fully specified word vocabularies, and perhaps more perniciously, a vast and complicated knowledge of the world that is not available in the training data. They also contain ambiguities and exhibit strong linguistic biases that allow a model to learn answering strategies that exploit those biases, without reasoning about the visual input [1, 15, 19].

To control for these issues, and to distill the core challenges of visual QA, the CLEVR visual QA dataset was developed [7]. CLEVR contains images of 3D-rendered objects, such as spheres and cylinders (Figure 2). Each image is associated with a number of questions that fall into different

categories. For example, `query attribute` questions may ask "*What is the color of the sphere?*", while `compare attribute` questions may ask "*Is the cube the same material as the cylinder?*".

For our purposes, an important feature of CLEVR is that many questions are explicitly relational in nature. Remarkably, powerful QA architectures [24] are unable to solve CLEVR, presumably because they cannot handle core relational aspects of the task. For example, as reported in the original paper a model comprised of ResNet-101 image embeddings with LSTM question processing and augmented with stacked attention modules vastly outperformed other models at an overall performance of $68.5\%$ (compared to $52.3\%$ for the next best, and $92.6\%$ human performance) [7]. However, for `compare attribute` and `count` questions (i.e., questions heavily involving relations across objects), the model performed little better than the simplest baseline, which answered questions solely based on the probability of answers in the training set for a given question category (Q-type baseline).

We used two versions of the CLEVR dataset: (i) the pixel version, in which images were represented in standard 2D pixel form. (ii) a state description version, in which images were explicitly represented by state description matrices containing factored object descriptions. Each row in the matrix contained the features of a single object – 3D coordinates (x, y, z); color (r, g, b); shape (cube, cylinder, etc.); material (rubber, metal, etc.); size (small, large, etc.). When we trained our models, we used *either* the pixel version or the state description version, depending on the experiment, but not both together.

## 3.2   Sort-of-CLEVR

To explore our hypothesis that the RN architecture is better suited to general relational reasoning as compared to more standard neural architectures, we constructed a dataset similar to CLEVR that we call "Sort-of-CLEVR"[2]. This dataset separates relational and non-relational questions.

Sort-of-CLEVR consists of images of 2D colored shapes along with questions and answers about the images. Each image has a total of 6 objects, where each object is a randomly chosen shape (square or circle). We used 6 colors (red, blue, green, orange, yellow, gray) to unambiguously identify each object. Questions are hard-coded as fixed-length binary strings to reduce the difficulty involved with natural language question-word processing, and thereby remove any confounding difficulty with language parsing. For each image we generated 10 relational questions and 10 non-relational questions. Examples of relational questions are: "*What is the shape of the object that is farthest from the gray object?*"; and "*How many objects have the same shape as the green object?*". Examples of non-relational questions are: "What is the shape of the *gray* object?"; and "*Is the blue object on the top or bottom of the scene?*". The dataset is also visually simple, reducing complexities involved in image processing.

## 3.3   bAbI

bAbI is a pure text-based QA dataset [22]. There are 20 tasks, each corresponding to a particular type of reasoning, such as deduction, induction, or counting. Each question is associated with a set of supporting facts. For example, the facts "*Sandra picked up the football*" and "*Sandra went to the office*" support the question "*Where is the football?*" (answer: "*office*"). A model succeeds on a task if its performance surpasses $95\%$. Many memory-augmented neural networks have reported impressive results on bAbI. When training jointly on all tasks using $10K$ examples per task, Memory Networks pass 14/20, DNC 16/20, Sparse DNC 19/20, and EntNet 16/20 (the authors of EntNets report state-of-the-art at 20/20; however, unlike previously reported results this was not done with joint training on all tasks, where they instead achieve 16/20) [23, 4, 17, 6].

## 3.4   Dynamic physical systems

We developed a dataset of simulated physical mass-spring systems using the MuJoCo physics engine [21]. Each scene contained 10 colored balls moving on a table-top surface. Some of the balls moved independently, free to collide with other balls and the barrier walls. Other randomly selected ball pairs were connected by invisible springs or a rigid constraint. These connections prevented the balls from moving independently, due to the force imposed through the connections. Input data consisted of state descriptions matrices, where each ball was represented as a row in a matrix with features

representing the rgb color values of each object and their spatial coordinates (x and y) across 16 sequential time steps.

The introduction of random links between balls created an evolving physical system with a variable number "systems" of connected balls (where "systems" refers to connected graphs with balls as nodes and connections between balls as edges). We defined two separate tasks: 1) infer the existence or absence of connections between balls when only observing their color and coordinate positions across multiple sequential frames, and 2) count the number of systems on the table-top, again when only observing each ball's color and coordinate position across multiple sequential frames.

Both of these tasks involve reasoning about the relative positions and velocities of the balls to infer whether they are moving independently, or whether their movement is somehow dependent on the movement of other balls through invisible connections. For example, if the distance between two balls remains similar across frames, then it can be inferred that there is a connection between them. The first task makes these inferences explicit, while the second task demands that this reasoning occur implicitly, which is much more difficult. For further information on all tasks, including videos of the dynamic systems, see the supplementary information.

## 4   Models

In their simplest form RNs operate on *objects*, and hence do not explicitly operate on images or natural language. A central contribution of this work is to demonstrate the flexibility with which relatively unstructured inputs, such as CNN or LSTM embeddings, can be considered as a set of objects for an RN. As we describe below, we require minimal oversight in factorizing the RN's input into a set of objects.

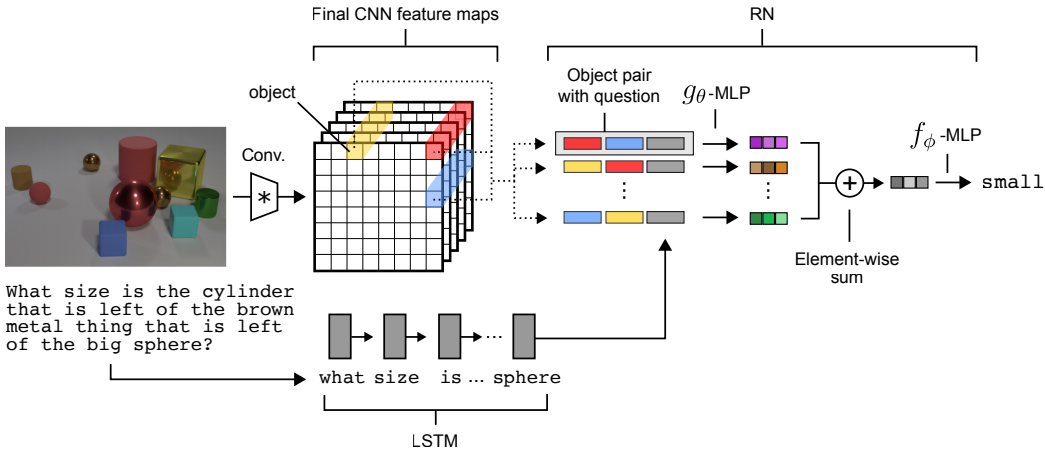

Figure 2: **Visual QA architecture**. Questions are processed with an LSTM to produce a question embedding, and images are processed with a CNN to produce a set of objects for the RN. Objects (three examples illustrated here in yellow, red, and blue) are constructed using feature-map vectors from the convolved image. The RN considers relations across all pairs of objects, conditioned on the question embedding, and integrates all these relations to answer the question.

**Dealing with pixels**   We used a CNN to parse pixel inputs into a set of objects. The CNN took images of size $128 \times 128$ and convolved them through four convolutional layers to $k$ feature maps of size $d \times d$, where $k$ is the number of kernels in the final convolutional layer. We remained agnostic as to what particular image features should constitute an object. So, after convolving the image, each of the $d^2$ $k$-dimensional cells in the $d \times d$ feature maps was tagged with a coordinate (from the range $(-1, 1)$ for each of the x- and y-coordinates)[3] indicating its relative spatial position, and was treated as an object for the RN (see Figure 2). This means that an "object" could comprise the background, a particular physical object, a texture, conjunctions of physical objects, etc., which affords the model great flexibility in the learning process.

**Conditioning RNs with question embeddings** The existence and meaning of an object-object relation should be question dependent. For example, if a question asks about a large sphere, then the relations between small cubes are probably irrelevant. So, we modified the RN architecture such that $g_\theta$ could condition its processing on the question: $a = f_\phi(\sum_{i,j} g_\theta(o_i, o_j, q))$. To get the question embedding $q$, we used the final state of an LSTM that processed question words. Question words were assigned unique integers, which were then used to index a learnable lookup table that provided embeddings to the LSTM. At each time-step, the LSTM received a single word embedding as input, according to the syntax of the English-encoded question.

**Dealing with state descriptions** We can provide state descriptions directly into the RN, since state descriptions are pre-factored object representations. Question processing can proceed as before: questions pass through an LSTM using a learnable lookup embedding for individual words, and the final state of the LSTM is concatenated to each object-pair.

**Dealing with natural language** For the bAbI suite of tasks the natural language inputs must be transformed into a set of objects. This is a distinctly different requirement from visual QA, where objects were defined as spatially distinct regions in convolved feature maps. So, we first took the 20 sentences in the support set that were immediately prior to the probe question. Then, we tagged these sentences with labels indicating their relative position in the support set, and processed each sentence word-by-word with an LSTM (with the same LSTM acting on each sentence independently). We note that this setup invokes minimal prior knowledge, in that we delineate objects as sentences, whereas previous bAbI models processed all word tokens from all support sentences sequentially. It's unclear how much of an advantage this prior knowledge provides, since period punctuation also unambiguously delineates sentences for the token-by-token processing models. The final state of the sentence-processing-LSTM is considered to be an object. Similar to visual QA, a separate LSTM produced a question embedding, which was appened to each object pair as input to the RN. Our model was trained on the joint version of bAbI (all 20 tasks simultaneously), using the full dataset of $10K$ examples per task.

**Model configuration details** For the CLEVR-from-pixels task we used: 4 convolutional layers each with 24 kernels, ReLU non-linearities, and batch normalization; 128 unit LSTM for question processing; 32 unit word-lookup embeddings; four-layer MLP consisting of 256 units per layer with ReLU non-linearities for $g_\theta$; and a three-layer MLP consisting of 256, 256 (with $50\%$ dropout), and 29 units with ReLU non-linearities for $f_\phi$. The final layer was a linear layer that produced logits for a softmax over the answer vocabulary. The softmax output was optimized with a cross-entropy loss function using the Adam optimizer with a learning rate of $2.5e^{-4}$. We used size 64 mini-batches and distributed training with 10 workers synchronously updating a central parameter server. The configurations for the other tasks are similar, and can be found in the supplementary information.

We'd like to emphasize the simplicity of our overall model architecture compared to the visual QA architectures used on CLEVR thus far, which use ResNet or VGG embeddings, sometimes with fine-tuning, very large LSTMs for language encoding, and further processing modules, such as stacked or iterative attention, or large fully connected layers (upwards of 4000 units, often) [7].

## 5 Results

### 5.1 CLEVR from pixels

Our model achieved state-of-the-art performance on CLEVR at $95.5\%$, exceeding the best model trained only on the pixel images and questions at the time of the dataset's publication by $27\%$, and surpassing human performance in the task (see Table 1 and Figure 3).

These results – in particular, those obtained in the `compare attribute` and `count` categories – are a testament to the ability of our model to do relational reasoning. In fact, it is in these categories that state-of-the-art models struggle most. Furthermore, the relative simplicity of the network components used in our model suggests that the difficulty of the CLEVR task lies in its relational reasoning demands, not on the language or the visual processing.

Many CLEVR questions involve computing and comparing more than one relation; for example, consider the question: *"There is a big thing on the right side of the big rubber cylinder that is behind*

| Model | Overall | Count | Exist | Compare Numbers | Query Attribute | Compare Attribute |
|---|---|---|---|---|---|---|
| Human | 92.6 | 86.7 | 96.6 | 86.5 | 95.0 | 96.0 |
| Q-type baseline | 41.8 | 34.6 | 50.2 | 51.0 | 36.0 | 51.3 |
| LSTM | 46.8 | 41.7 | 61.1 | 69.8 | 36.8 | 51.8 |
| CNN+LSTM | 52.3 | 43.7 | 65.2 | 67.1 | 49.3 | 53.0 |
| CNN+LSTM+SA | 68.5 | 52.2 | 71.1 | 73.5 | 85.3 | 52.3 |
| CNN+LSTM+SA* | 76.6 | 64.4 | 82.7 | 77.4 | 82.6 | 75.4 |
| CNN+LSTM+RN | **95.5** | **90.1** | **97.8** | **93.6** | **97.9** | **97.1** |

\* Our implementation, with optimized hyperparameters and trained end-to-end using the same CNN as in our RN model. We also tagged coordinates, which did not improve performance.

Table 1: **Results on CLEVR from pixels.** Performances of our model (RN) and previously reported models [8], measured as accuracy on the test set and broken down by question category.

*the large cylinder to the right of the tiny yellow rubber thing; What is its shape?"*, which has three spatial relations ("right side", "behind", "right of"). On such questions, our model achieves 93% performance, indicating that the model can handle complex relational reasoning.

**Results using privileged training information**   A more recent study reports overall performance of 96.9% on CLEVR, but uses additional supervisory signals on the functional programs used to generate the CLEVR questions [8]. It is not possible for us to directly compare this to our work since we do not use these additional supervision signals. Nonetheless, our approach greatly outperforms a version of their model that was not trained with these extra signals, and even a version of their model trained using $9K$ ground-truth programs. Thus, RNs can achieve very competitive, and even super-human results under much weaker and more natural assumptions, and even in situations when functional programs are unavailable.

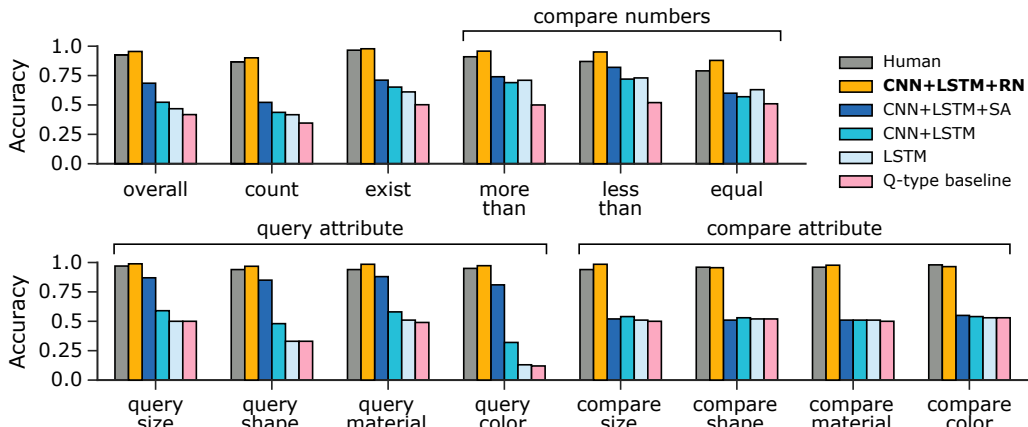

Figure 3: **Results on CLEVR from pixels.** The RN-augmented model outperformed all other models and exhibited super-human performance overall. In particular, it solved "compare attribute" questions, which trouble all other models because they heavily depend on relational reasoning.

## 5.2   CLEVR from state descriptions

To demonstrate that the RN is robust to the form of its input, we trained our model on the state description matrix version of the CLEVR dataset. The model achieved an accuracy of 96.4%. This result demonstrates the generality of the RN module, showing its capacity to learn and reason about object relations while being agnostic to the kind of inputs it receives – i.e., to the particular representation of the object features to which it has access. Therefore, RNs are not necessarily

restricted to visual problems, and can thus be applied in very different contexts, and to different tasks that require relational reasoning.

## 5.3  Sort-of-CLEVR from pixels

The results so far led us to hypothesize that the difficulty in solving CLEVR lies in its heavy emphasis on relational reasoning, contrary to previous claims that the difficulty lies in question parsing [9]. However, the questions in the CLEVR dataset are not categorized based on the degree to which they may be relational, making it hard to assess our hypothesis. Therefore, we use the Sort-of-CLEVR dataset which we explicitly designed to seperate out relational and non-relational questions (see Section 3.2).

We find that a CNN augmented with an RN achieves an accuracy above $94\%$ for both relational and non-relational questions. However, a CNN augmented with an MLP only reached this performance on the non-relational questions, plateauing at $63\%$ on the relational questions. This strongly indicates that models lacking a dedicated relational reasoning component struggle, or may even be completely incapable of solving tasks that require very simple relational reasoning. Augmenting these models with a relational module, like the RN, is sufficient to overcome this hurdle.

A simple "closest-to" or "furthest-from" relation is particularly revealing of a CNN+MLP's lack of general reasoning capabilities ($52.3\%$ success). For these relations a model must gauge the distances between *each* object, and then compare each of these distances. Moreover, depending on the images, the relevant distance could be quite small in magnitude, or quite large, further increasing the combinatoric difficulty of this task.

## 5.4  bAbI

Our model succeeded on $18/20$ tasks. Notably, it succeeded on the basic induction task ($2.1\%$ total error), which proved difficult for the Sparse DNC ($54\%$), DNC ($55.1\%$), and EntNet ($52.1\%$). Also, our model did not catastrophically fail in any of the tasks: for the 2 tasks that it failed (the "two supporting facts", and "three supporting facts" tasks), it missed the $95\%$ threshold by $3.1\%$ and $11.5\%$, respectively. We also note that the model we evaluated was chosen based on overall performance on a withheld validation set, using a single seed. That is, we did not run multiple replicas with the best hyperparameter settings (as was done in other models, such as the Sparse DNC, which demonstrated performance fluctuations with a standard deviation of more than $\pm 3$ tasks passed for the best choice of hyperparameters).

## 5.5  Dynamic physical systems

Finally, we trained our model on two tasks requiring reasoning about the dynamics of balls moving along a surface. In the connection inference task, our model correctly classified all the connections in $93\%$ of the sample scenes in the test set. In the counting task, the RN achieved similar performance, reporting the correct number of connected systems for $95\%$ of the test scene samples. In comparison, an MLP with comparable number of parameters was unable to perform better than chance for both tasks. Moreover, using this task to learn to infer relations results in transfer to unseen motion capture data, where RNs predict the connections between body joints of a walking human (see Supplementary Material for experimental details and example videos).

## 6  Discussion and Conclusions

RNs are powerful, versatile, and simple neural network modules with the capacity for relational reasoning. The performance of RN-augmented networks on CLEVR is especially notable; they significantly improve upon current general purpose, state-of-the-art models (upwards of $25\%$), indicating that previous architectures lacked a fundamental, general capacity to reason about relations. Moreover, these results unveil an important distinction between the often confounded notions of *processing* and *reasoning*. Powerful visual QA architectures contain components, such as ResNets, which are highly capable visual processors capable of detecting complicated textures and forms. However, as demonstrated by CLEVR, they lack an ability to *reason* about the features they detect.

RNs can easily exploit foreknowledge of the relations that should be computed for a particular task. Indeed, especially in circumstances with strong computational constraints, bounding the otherwise quadratic complexity of the number of relations could be advantageous. Attentional mechanisms could reduce the number of objects fed as input to the RN, and hence reduce the number of relations that need to be considered. Or, using an additional down-sampling convolutional or pooling layer could further reduce the number of objects provided as input to the RN; indeed, max-pooling to $4 \times 4$ feature maps reduces the total number of objects, and hence computed relations, and results in $87\%$ performance on the validation set.

RNs have a flexible input format: a set of objects. Our results show that, strikingly, the set of objects does not need to be cleverly pre-factored. RNs learn to deal with "object" representations provided by CNNs and LSTMs, presumably by influencing the content and form of the object representations via the gradients they propagate.

In future work it would be interesting to apply RNs to relational reasoning across highly abstract entities (for example, decisions in hierarchical reinforcement learning tasks). Relation reasoning is a central component of generally intelligent behavior, and so, we expect the RN to be a simple-to-use, useful and widely used neural module.

**Acknowledgments**

We would like to thank Murray Shanahan, Ari Morcos, Scott Reed, Daan Wierstra, and many others on the DeepMind team, for critical feedback and discussions.

## Footnotes

[2]The "Sort-of-CLEVR" dataset will be made publicly available online

[3]We also experimented without this tagging, and achieved performance of $88\%$ on the validation set.

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
