[Supplementary Material]

# Supplementary Material:
# A simple neural network module
# for relational reasoning

Here we provide additional details on related work, CLEVR from pixels, CLEVR from state descriptions, Sort-of-CLEVR, bAbI, and Dynamic physical system reasoning. For each task, we provide additional information on the dataset, model architecture, training and results where necessary.

## 1   Related Work

Since the RN is highly versatile, it can be used for visual, text-based, and state-based tasks. As such, it touches upon a broad range of areas in machine learning, computer vision, and natural language understanding. Here, we provide a brief overview of some of the most relevant related work.

### 1.1   Relational reasoning

Relational reasoning is implicit in many symbolic approaches [9, 25] and has been explicitly pursued using neural networks as well [4]. There is recent work applying neural networks to graphs, which are a natural structure for formalising relations [10, 14, 26, 29, 19, 2]. Perhaps a crucial difference between this work and our work here is that RNs require minimal oversight to produce their input (a set of objects), and can be applied successfully to tasks even when provided with relatively unstructured inputs coming from CNNs and LSTMs. There has also been some recent work on reasoning about sets, although this work does not explicitly reason about the relations of elements *within* sets [38].

### 1.2   Grounding spatial relations

Although grounding language in spatial percepts has a long-standing tradition, the majority of previous research has focused on either rule-based spatial representations or hand-engineered spatial features [7, 8, 15, 16, 18, 22, 30, 31]. Although there are some attempts to learn spatial relations using spatial templates [21, 23], these approaches are less versatile than ours.

### 1.3   Visual question answering

Visual question answering is a recently introduced task that measures a machine understanding of the scene through questions [1, 22]. Related to our work, we are mostly interested in the newly introduced CLEVR dataset [12] that distills core challenges of the task, namely relational and multi-modal reasoning. The majority of approaches to question answering share the same pipeline [6, 24, 28]. First, questions are encoded with recurrent neural networks, and images are encoded with convolutional neural networks. Next, both representations are combined, and the answers are either predicted or generated. Most successful methods also use an attention mechanism that locate important image regions [5, 35, 36, 37]. In our work, we follow a similar pipeline, but we use Relation Networks as a powerful reasoning module.

Parallel to our work, two architectures have shown impressive results on the CLEVR dataset [11, 13]. Both approaches hinge on compositionality principles, and have shown they are capable of some relational reasoning. However, both require either designing modules, or require direct access to

ground-truth programs. The RN module, on the other hand, is conceptually simpler, can readily be combined with basic neural components such as CNNs or LSTMs, can be broadly applied to various tasks, and achieves significantly better results on CLEVR [12] than [11], and on par with strongly supervised system of [13].

## 1.4 Text-based question answering

Answering text-based questions has long been an active research area in the NLP community [3, 17, 20, 39]. Recently, in addition to traditional symbolic-based question answering architectures, we observe a growing interest in neural-based approaches to text based question answering [27, 33, 34]. While these architectures rely on 'memories', we empirically show that the RN module has similar capabilities, reaching very competitive results on the bAbI dataset [32] – a dataset that test reasoning capabilities of text-based question answering models.

## 2 CLEVR from pixels

Our model (described in Section 4 of the main text) was trained on 70,000 scenes from the CLEVR dataset and a total of 699,989 questions. Images were first down-sampled to size $128 \times 128$, then pre-processed with padding to size $136 \times 136$, followed by random cropping back to size $128 \times 128$ and slight random rotations between $-0.05$ and $0.05$ rads. We used 10 distributed workers that synchronously updated a central parameter server. Each worker learned with mini-batches of size 64, using the Adam optimizer and a learning rate of $2.5e^{-4}$. Dropout of $50\%$ was used on the penultimate layer of the RN. In our best performing model each convolutional layer used 24 kernels of size $3 \times 3$ and stride 2, batch normalization, and rectified linear units. The model stopped improving in performance after approximately 1.4 million iterations, at which point training was concluded. The model achieved $96.8\%$ accuracy on the validation set. In general, we found that smaller models performed best. For example, 128 hidden unit LSTMs performed better than 256 or 512, and CNNs with 24 kernels were better than CNNs with more kernels, such as 32, 64, or more.

### 2.1 Results: failure cases

Although our model gets most answers correct, a closer examination of the failure cases help us to identify limitations of our architecture. In Table 1 (supplementary material), we show some examples of CLEVR questions that our model fails to answer correctly, along with the ground-truth answers. Based on our observations, we hypothesize that our architecture fails especially when objects are heavily occluded, or whenever a high precision object position representation is required. We also observe that many failure cases for our model are also challenging for humans.

## 3 CLEVR from state descriptions

The model that we train on the state description version of CLEVR is similar to the model trained on the pixel version of CLEVR, but without the vision processing module. We used a 256 unit LSTM for question processing and word-lookup embeddings of size 32. For the RN we used a four-layer MLP with 512 units per layer, with ReLU non-linearities for $g_\theta$. A three-layer MLP consisting of 512, 1024 (with $2\%$ dropout) and 29 units with ReLU non-linearities was used for $f_\theta$. To train the model we used 10 distributed workers that synchronously updated a central parameter server. Each worker learned with mini-batches of size 64, using the Adam optimizer and a learning rate of $1e^{-4}$.

## 4 Sort-of-CLEVR

The Sort-of-CLEVR dataset contains 10,000 images of size $75 \times 75$, of which 200 were withheld for validation. There were 20 questions generated per image (10 relational and 10 non-relational).

Non-relational questions are split into three categories: (i) query shape, e.g. "*What is the shape of the red object?*"; (ii) query horizontal position, e.g. "*Is the red object on the left or right of the image?*"; (iii) query vertical position, e.g. "*Is the red object on the top or bottom of the image?*". These questions are non-relational because one can answer them by reasoning about the attributes (e.g. position, shape) of a single entity which is identified by its unique color (e.g. *red*).

Relational questions are split into three categories: (i) closest-to, e.g. "*What is the shape of the object that is closest to the green object?*"; (ii) furthest-from, e.g. "*What is the shape of the object that is furthest from the green object?*"; (iii) count, e.g. "*How many objects have the shape of the green object?*". We consider these relational because answering them requires reasoning about the attributes of one or more objects that are defined relative to the attributes of a reference object. This reference object is uniquely identified by its color.

Questions were encoded as binary strings of length 11, where the first 6 bits identified the color of the object to which the question referred, as a one-hot vector, and the last 5 bits identified the question type and subtype.

In this task our model used: four convolutional layers with 32, 64, 128 and 256 kernels, ReLU non-linearities, and batch normalization; the questions, which were encoded as fixed-length binary strings, were treated as question embeddings and passed directly to the RN alongside the object pairs; a four-layer MLP consisting of 2000 units per layer with ReLU non-linearities was used for $g_\theta$; and a four-layer MLP consisting of 2000, 1000, 500, and 100 units with ReLU non-linearities used for $f_\phi$. An additional final linear layer produced logits for a softmax over the possible answers. The softmax output was optimized with a cross-entropy loss function using the Adam optimizer with a learning rate of $1e^{-4}$ and mini-batches of size 64.

We also trained a comparable MLP based model (CNN+MLP model) on the Sort-of-CLEVR task, to explore the extent to which a standard model can learn to answer relational questions. We used the same CNN and LSTM, trained end-to-end, as described above. However, this time we replaced the RN with an MLP with the same number of layers and number of units per layer. Note that there are more parameters in this model because the input layer of the MLP connects to the full CNN image embedding.

## 5 bAbI model for language understanding

For the bAbI task, each of the 20 sentences in the support set was processed through a 32 unit LSTM to produce an object. For the RN, $g_\theta$ was a four-layer MLP consisting of 256 units per layer. For $f_\phi$, we used a three-layer MLP consisting of 256, 512, and 159 units, where the final layer was a linear layer that produced logits for a softmax over the answer vocabulary. A separate LSTM with 32 units was used to process the question. The softmax output was optimized with a cross-entropy loss function using the Adam optimizer with a learning rate of $2e^{-4}$.

## 6 Dynamic physical system reasoning

For the connection inference task the targets were binary vectors representing the existence (or non-existence) of a connection between each ball pair. For a total of 10 objects, the targets were $10^2$ length vectors. For the counting task, the targets were one-hot vectors (of length 10) indicating the number of systems of connected balls. It is important to point out that in the first task the supervision signal provided by the targets explicitly informs about the relations that need to be computed. In the second task, the supervision signal (counts of systems) do not provide explicit information about the kind of relations that need to be computed. Therefore, the models that solve the counting task must successfully infer the relations implicitly.

Inputs to the RN were state descriptions. Each row of a state description matrix provided information about a particular object (i.e. ball), including its coordinate position and color. Since the system was dynamic, and hence evolved through time, each row contained object property descriptions for 16 consecutive time-frames. For example, a row could be comprised of 33 floats: 16 for the object's $x$ coordinate position across 16 frames, 16 for the object's $y$ coordinate position across 16 frames, and 1 for the object's color. The RN treated each row in this state description matrix as an object. Thus, it had to infer an object description contained information of the object's properties evolving through time.

For the connection inference task, the RN's $g_\theta$ was a four-layer MLP consisting of three layers with 1000 units and one layer with 500 units. For $f_\phi$, we used a three-layer MLP consisting of 500, 100, and 100 units, where the final layer was a linear layer that produced logits corresponding to the existence/absence of a connection between each ball pair. The output was optimized with a

**Non-relational question**

Q: What is the shape of the gray object?
A: circle

**Relational question**

Q: What is the shape of the object
that is furthest from the gray object?
A: square

Image

**Non-relational question**

Q: Is the green object on the left or on the right?
A: right

**Relational question**

Q: How many objects have the shape of the orange object?
A: 3

**Non-relational question**

Q: Is the yellow object on the top or on the bottom?
A: bottom

**Relational question**

Q: What is the color of the object that is closest to the blue object?
A: red

**Non-relational question**

Q: What is the shape of the red object?
A: circle

**Relational question**

Q: How many objects have the shape of the blue object?
A: 1

**Non-relational question**

Q: Is the blue object on the top or on the bottom?
A: top

**Relational question**

Q: What is the shape of the object that is closest to the red object?
A: yellow

Figure 1: **"Sort-of-CLEVR" task: examples and results.** The Sort-of-CLEVR example here consists of an image of six objects and two questions – a relational question, and a non-relational question – along with the corresponding answers. The fraction of correctly answered relational questions (inset bar plot) for our model (CNN+RN) is much larger than the comparable MLP based model (CNN+MLP), whereas both models have similar performance levels for non-relational questions.

cross-entropy loss function using the Adam optimizer with a learning rate of $1e^{-4}$ and a batch size of 50. The same model was used for the counting task, but this time the output layer of the RN was a linear layer with 10 units. For baseline comparisons we replaced the RNs with MLPs with comparable number of parameters.

Please see the supplementary videos.

| What shape is the small object that is in front of the yellow matte thing and behind the gray sphere? | What number of things are either tiny green rubber objects or shiny things that are behind the big metal block? | What number of objects are blocks that are in front of the large red cube or green balls? |
|---|---|---|
| *RN:*    cylinder | 1 | 2 |
| *GT:*    cube | 2 | 3 |

| Is the shape of the small red object the same as the large matte object that is right of the small rubber ball? | How many gray objects are in front of the tiny green shiny ball and right of the big blue matte thing? | What number of objects are big red matte cubes or things on the right side of the large red matte block? |
|---|---|---|
| *RN:*    no | 0 | 5 |
| *GT:*    yes | 1 | 6 |

| There is a brown ball; what number of things are left of it? | How many objects are big purple rubber blocks or red blocks in front of the tiny yellow rubber thing? | How many things are rubber cylinders in front of the tiny yellow block or blocks that are to the right of the small brown rubber thing? |
|---|---|---|
| *RN:*    3 | 3 | 2 |
| *GT:*    4 | 2 | 3 |

| What number of objects are either big things that are left of the cylinder or cylinders? | Are there the same number of small blue objects that are to the right of the blue cube and blue metal cubes? | What number of other things are there of the same material as the green cube? |
|---|---|---|
| *RN:*    2 | no | 6 |
| *GT:*    3 | yes | 5 |

Table 1: Failures on CLEVR; RN – predicted answers, GT – ground-truth answer.