[Reviews · NeurIPS 2017]

Reviewer 1



The paper proposes a plug and play module (called Relation Networks (RNs)) specialized for relational reasoning. The module is composed of Multi Layer Perceptrons and considers relations between all pairs of objects. The proposed module when plugged into traditional networks achieves state of the art performance on the CLEVR visual question answering dataset, state of the art (with joint training for all tasks) on the bAbI textual question answering dataset and high performance (93% on one task and 95% on another) on a newly collected dataset of simulated physical mass-spring systems. The paper also collects a dataset similar to CLEVR to demonstrate the effectiveness of the proposed RNs for relational questions. Strengths: 1. The proposed Relation Network is a novel neural network specialized for relational reasoning. The success of the proposed network is extensively shown by experimenting with three different tasks and clearly analyzing the effectiveness for relational questions by collecting a novel dataset similar to CLEVR. 2. The proposed RNs have been shown to be able to work with different forms of input -- explicit state representations as well as features from a CNN or LSTM. 3. The paper is well written and the details of model architecture including hyperparameters are provided. 4. As argued in the paper, I agree that relational reasoning is central to intelligence and since RNs are shown to be able to achieve this reasoning and a result perform better at tasks requiring such reasoning than existing networks, RNs seem to be of significant importance for designing reasoning networks. Weaknesses: 1. Could authors please analyze and comment on how complicated relations can be handled by RNs. Is it the case that RNs perform well for single hop relations such as "what is the color of the object closest to the blue object" which requires reasoning about only one hop relation (distance between blue object and all other objects), but not so well for multiple hop relations such as "What shape is the small object that is in front of the yellow matte thing and behind the gray sphere?". From the failure cases in table 1 of supplementary material, it seems that the model has difficulty in answering questions involving multiple hops of relations. 2. L203-204, it is not clear to me what do authors mean by "we tagged ... support set". Is this referring to some form of human annotation? If so, could authors please elaborate on what happens at test time? 3. All the datasets experimented with in the paper are synthetic datasets. Could authors please comment on how they expect the RNs to work on real datasets such as the VQA dataset from Antol et al.? Post-rebuttal comments: Authors have provided satisfactory response to my question about multi-hop reasoning. However, I would still like to see experiments on real VQA dataset to see how effective RNs are at dealing with the amount of variation real datapoints show (in vision as well as in language). So it would be great if authors could include results on the VQA dataset (Antol et al., ICCV 2015) in camera-ready.

Reviewer 2



This paper presents the relational network module, which when included as part of a larger network architecture is able to essentially solve the CLEVR VQA task despite its simplicity. The model is also tested over other synthetic tasks involving both vision and language; I hope that in the future the authors can also demonstrate its effectiveness on real-world tasks. While at this point I don't think results on bAbI are particularly informative, especially with the strange way the bAbI task is set up in this paper (what does "up to 20 support sentences" mean? was there some rule-based or ML method used to select these support sentences? why not use all sentences as support?), the model is an interesting and effective way to think about relational QA problems, and I hope that the paper is accepted. That said, I have some questions/comments about the paper: - were functions other than simple summation in Eq.1 experimented with? - what is the function of the "arbitrary coordinate" in the "dealing with pixels" section? How is it "arbitrary" if it is supposed to indicate relative position? there is not enough detail to understand how it is implemented, and Figure 2 offers no insight despite being referenced. Is this crucial to making the RN work? If so, it needs to be stated in the paper. - is the reason why all support sentences aren't used in bAbI due to computational constraints (since some of the bAbI tasks have large contexts), and is this a potential limitation of the RN? - the model doesn't make as much sense for NLP tasks where "objects" aren't clearly defined, and there is an obvious order to the objects (there is discourse-level information in the sequential ordering of sentences, even in bAbI). Couldn't the model be applied across different units of text (e.g., treat words, phrases, sentences, paragraphs all as objects)? Any thoughts on how this could be implemented in for example bAbI? - the CNN used in the CLEVR experiments is very simple compared to prior work, which utilized VGG features. since the CLEVR images are artificially constructed, it strikes me that a simpler CNN is better suited for the problem. What happens if you run the RN on CLEVR but utilize feature maps from the last conv layer of VGG as objects? this would be a more fair comparison than what is described in the paper.